# Strategies for Improving the Utilization of Preventive Care Services: Application of Importance–Performance Gap Analysis Method

**DOI:** 10.3390/ijerph192013195

**Published:** 2022-10-13

**Authors:** Ming-Jye Wang, Yi-Ting Lo

**Affiliations:** 1Department of Secretariat, National Taiwan University Hospital Hsin-Chu Branch, No. 25, Lane 442, Sec. 1, Jingguo Rd., Hsinchu City 300, Taiwan; 2Department of Development and Planning, National Taiwan University Hospital Hsin-Chu Branch, Hsinchu City 300, Taiwan

**Keywords:** importance–performance gap analysis, preventive care service, health check-ups

## Abstract

The utilization of preventive care services is limited. Previous studies based on communities have looked at many types of barriers to preventive care (i.e., why not do). This study aimed to gain an insight into the key factors and performance gaps (i.e., why do and how to do) of people who attended a regional teaching hospital to undergo health check-ups using a purposive sampling method to recruit people to complete a self-administered questionnaire. Paired sample *t*-tests and importance–performance gap and regression analyses were performed. The results indicated that the participants’ wish to understand their health status, the quality of medical devices and the completeness of items, and process layout planning were the key factors that affect people’s health check-up behavior. Promoting the effectiveness of hospital performance may improve the positive cycle of people’s health check-up behavior from the community to the hospital. Continuing to promote the knowledge of preventive care services is necessary, and it is very important for people to have a good experience of undergoing health check-ups in the hospital. Important strategies for improving the utilization of preventive care services may include: meeting the person’s personalization needs, improving the quality of medical devices and the completeness of items, and having appropriate process layout planning, a detailed interpretation of the results, and further follow-ups in the hospital.

## 1. Introduction

Good health outcomes come not only from receiving high-quality medical care when it is needed, but also from the early detection and prevention of health problems [1]. Therefore, the goal of Healthy People 2030 is to “help people get recommended preventive health care services” [2]. Disease prevention has become a key consideration for public health policymakers around the world. The promotion of health programs focuses more on health promotion and disease prevention in order to reduce the burden of disease and associated risk factors [3]. Transforming health from disease treatment to disease prevention and health management, and from passive to active and predictive care [4], is the trend of global health progress. Studies have evidenced that 60% of diseases are caused by unhealthy lifestyles [5] and, in fact, these unhealthy lifestyles which are related to diseases can be prevented and controlled [4]. Clinical prevention strategies include intervention before the disease occurs (primary prevention), the early detection and treatment of disease stages (secondary prevention), and the control of the disease to slow or stop its progression (tertiary prevention) [6]. These interventions combined with lifestyle changes can greatly reduce the incidence of chronic diseases and their associated disability and death [7]. According to the theory of health economics, investment in health capital, such as the use of preventive care services, can reduce the incidence of disease and death [8], effectively improve and maintain personal health, reduce the utilization or expenditure of medical care [9,10,11], help promote a healthier lifestyle, detect disease earlier, and reduce the need for an individual’s inpatient medical care [11]. Therefore, more and better medical services cannot solve declining health conditions or slow the rise in medical costs [12]. Promoting preventive care services is an effective strategy to reduce the need for medical services and expenditures.

However, the utilization of preventive care services is very low; even for free services in the United States, only one in four middle-aged and older adults (50–64 years old) and one in two seniors (65 years and older) receive the recommended preventive care services [13,14]. In China, the utilization of preventive care services is even lower, with utilization among adults being less than 7% [15]. In Taiwan, the average annual utilization rate of adult preventive health services was about 30% in 2018 [16]. Cancer is the leading cause of death worldwide and approximately 40% of cancers are preventable through lifestyle changes and screening [17]. Despite the use of various programs to encourage people to undergo cancer screening, the rates of participation are consistently below what is expected in many countries [18], so exploring how to promote preventive care services is an important issue in changing health status.

Changes in the utilization of clinical preventive services in developed countries are mainly attributable to health insurance and the social determinants of health (socioeconomic factors, health-related behaviors, and environmental factors). However, there are still differences in the use of clinical preventive services among older adults with health insurance coverage after removing their associated structural and financial barriers [19]. It can be seen that health insurance and social determinants of health are no longer sufficient to explain the low utilization of preventive care services, and thus, some scholars have focused on the determinants of demand and the issue of equal access to preventive care services [20,21,22]. In addition, the fragmentation of health planning and the difficulty in identifying the performance of the roles of different behaviors involved in preventive care services are also factors [23,24]. From a methodological view, these may be yet-to-be-established frameworks for measuring and evaluating preventive care [3], since preventive care services involve the strategic management issues of providers, customers, quality, efficiency, and resource allocation. The two-dimensional strategy matrix (importance–performance gap analysis, IPGA) [25] developed from the perspective of customers has been applied in various fields because it can quickly provide decision makers with useful information. The matrix vertical axis is relative importance (people’s perception of importance) while the horizontal axis is relative performance (people’s behavior). Therefore, the research hypotheses are that the greater the people’s perception of importance, the greater their perception of behavior performance as well. Otherwise, there will be gaps, and it is urgent to review and improve these gaps.

This study attempted to use this management method to construct effective and feasible related strategies. The aims of this study were: (1) to explore the key factors of health check-up preventive behavior of people in hospitals; (2) to analyze the gaps between people’s perceived importance and behavioral performance in health check-up preventive behavior in hospitals; and (3) to examine the relationship between people’s characteristics, health status, and performance gaps. The findings of this study may provide reference for health policies, help improve the utilization of preventive care services, control medical costs, and achieve the goals of personal health and well-being.

## 2. Methods

### 2.1. Study Participants

The study participants were people who attended a regional teaching hospital in Hsinchu City, Taiwan, for company employee health check-ups, individual private health check-ups, and adult preventive health services. A purposive sampling method was used to recruit people with consent from 30 July to 14 September 2020. A total of 476 completed self-administered questionnaires were collected. 

Company employee health check-ups refers to Article 20, the Occupational Safety and Health Act in Taiwan: “the employers shall conduct pre-employment physical examinations for laborers at the time of employment; for currently employed laborers, the following health examinations shall be conducted: 1. General health examinations; 2. Special health examinations for those involved in tasks with special health hazards; 3. Health examinations of specific items for specific targets workers as designated by the central competent authority” [26]. Individual private health check-ups refer to individuals making their own decision to voluntarily go to a hospital for a health check-up. Adult preventive health services refer to the National Health Insurance in Taiwan: “the government provided preventive healthcare service for adults, including physical examinations, blood and urine tests, and health consultations. These are provided free of charge to people aged 40–64 every three years, and to those aged 65 or over every year” [27].

### 2.2. Research Scale Design

The questionnaire for this study was developed with reference to a literature review to collect the 14 influencing factors related to health check-ups, and the draft was reviewed through a discussion with an expert panel, including a family physician, nurse, and administrator. The questionnaire items were examined for applicability and could be easily administered. The contents of the questionnaire included three dimensions: influencing factors, health status, and people’s characteristics. There were 14 influencing factors (items), which were: (1) sufficient health knowledge [28,29,30], (2) motivated by friends and family [31,32], (3) with a companion [32], (4) a wish to understand one’s health status [32], (5) previous experience (pleasant or unpleasant) [30,33], (6) the completeness of items [32], (7) an expense needed [32,34], (8) a convenient location [30,32], (9) the quality of a medical device [30,32], (10) process layout planning [30], (11) whether an incentive was offered (such as gifts and discounts) [32], (12) being asymptomatic without a check [30,35], (13) a lack of time (at work or taking care of children) [35,36], and (14) being afraid of potential health problems [30,33]. Each item was scored based on people’s perception of its importance (1 = very unimportant to 5 = very important) and the level of influence on people’s behavior (i.e., behavioral performance of health check-up preventive behavior in hospitals, 1 = always influenced to 5 = not influenced at all). 

Health status included the following: whether they had received medical treatment in the past year, their BMI, whether they had a history of chronic disease, and whether they had undergone health screening in the past 3 years. 

People’s characteristics included gender, age, education, marital status, and occupation. 

### 2.3. Data Analysis

Descriptive statistics, such as the mean, standard deviation, frequency, and rank order, were used to investigate the key factors of the health check-up preventive behavior of people in the hospital. A comparison of the differences in demographic characteristics was conducted using an ANOVA and a chi-square test. 

Paired sample *t*-tests were used to analyze the gaps between people’s perceived importance and behavioral performance of health check-up preventive behavior in hospitals. Factors with statistical significance (*p* < 0.05) were defined as performance gaps. Gaps between people’s perceived importance and behavioral performance are described briefly as gaps between perception and performance in the full text.

Regression analysis was used to examine the relationship between people’s characteristics, health status, and performance gaps. 

All statistics were analyzed using SPSS version 22.0.

The IPGA matrix was used to develop people-centric health check-up strategic management. The IPGA tool is a simple graphical tool enabling the comparison of perceived importance against performance, which is expected to help make service decisions through a simple strategic matrix. Furthermore, it includes gap theory, enabling the identification of service failures to be based on the user’s expectation against their perception of the provided services. Therefore, understanding people demand will provide an effective management for people’s health check-up preventive behavior in hospitals. The IPGA model includes the following six steps:

Step 1: Collect people’s perceptions of importance and behavioral performance of health check-up preventive behavior in hospitals on all 14 items of the influencing factors. 

Step 2: Calculate the average importance value of each item ($\bar{I}$*_j_*), the average performance value of each item ($\bar{P}$*_j_*), the average importance value of all items ($\bar{I}$), and the average performance value of all items ($\bar{P}$).

Step 3: Use paired sample *t*-tests to analyze whether the gap between the perception and performance for each item of health check-up influencing factors is a positive gap (performance > importance) or a negative gap (performance < importance), or whether there is not a gap (performance = importance).

Step 4: Compute the relative importance (RI) and relative performance (RP), RI = ($\bar{I}$*_j_/*$\bar{I})$. If $\bar{P_{j}}>\bar{I_{j}}$ and the *t*-test is significant, RP (*j*) = $\bar{P}$*_j_/*$\bar{P}$. If $\bar{P_{j}}<\bar{I_{j}}$ and the *t*-test is significant, RP (*j*) = −($\bar{P}$*_j_/*$\bar{P})$^−1^. If $\bar{P_{j}}>\bar{I_{j}}$ or $\bar{P_{j}}<\bar{I_{j}}$ and the *t*-test is non-significant, RP (*j*) = 0.

Step 5: Draw the IPGA strategic matrix (Figure 1), where relative importance (people’s perception of importance) is the vertical axis, relative performance (people’s behavior) is the horizontal axis, and the intersection is fixed at (0,1). The IPGA grid represents the different strategies for resource allocation and management, as illustrated below.

(1) Quadrant I is composed of a high relative performance and a high relative importance and corresponds to “Keep up the good work”. (2) Quadrant II is composed of a low relative performance and a high relative importance and corresponds to “Concentrate here”. A point further away from coordinate (0,1) indicates a greater need for improvement. (3) Quadrant III is composed of a low relative performance and a low relative importance and corresponds to “Low priority”. (4) Quadrant IV is composed of a high relative performance and a low relative importance and corresponds to “Possible overkill”. A point further away from coordinate (0,1) indicates a greater need to re-allocate resources. 

Step 6: Determine the priorities for resource allocation for the items in Quadrant II.

The distance D(*j*) indicates the priority for improvement:$$D_{q}\left(j\right)=\sqrt{[{\bar{P}}_{j}/\underset{r\in q}{\max}{\left(|{\bar{P}}_{.r}|\right]}^{2}+{\left[\left({\bar{I}}_{.j}-1\right)/\underset{r\in q}{\max}(|{\bar{I}}_{.r}-1|)\right]}^{2}}$$

## 3. Results

### 3.1. Study Participant Characteristics and Comparative Analysis

Among the 476 participants, 266 came for company employee health check-ups, 108 came for individual private health check-ups, and 102 came for adult preventive health services. Through an ANOVA and the chi-square test, there were significant differences in age, gender, education, marital status, occupation, experience in medical treatment, and history of chronic disease. The group of company employee health check-ups was relatively young, with an average age of 39.9 years old. Most of them were men, accounting for 68.4%, and had a high education level, with 93.6% having a college degree or above, and 89.5% had no history of chronic diseases. Women were the majority in the individual health check-up and adult preventive health service groups, accounting for 57.4% and 54.9%, respectively, and of those, more than ¾ were married. The adult preventive health service group had a lower education level, where 41.2% of them had a high school education or below, and more people who had not worked, accounting for 42.2%, and about ½ (51.5%) had experience in medical treatment in the past 1 year. More than ¾ (89.5%, 80.6%, and 76.2%) of the participants had no history of chronic disease (Table 1).

### 3.2. Key Factors Affecting Health Check-Up Behavior

For the 14 items of influencing factors perceived as important and having an influence on health check-up behaviors as seen by the people, Cronbach’s *α* value was 0.78 and 0.86, respectively. If the people’s perception of being important and very important was classified as a “positive” factor, the higher the percentage, the more people that thought the factor was important. If frequently influenced and always influenced were classified as “negative” factors, the higher the percentage, the more the people’s behavior was affected by these factors, and the worse the performance. The rankings are shown in Table 2. A wish to understand one’s health status and the quality of a medical device were the key factors that affect health check-up behavior (high positive % and negative %, ranking 1–3), and even the completeness of items was an important factor for company employee health check-ups and individual private health check-ups (ranking 3–4). Process layout planning was especially critical in adult preventive health services (negative %, ranking 2). However, the factors of incentives offered (such as gifts and discounts) and being asymptomatic without a check were not considered important to the people and did not affect their health check-up behavior.

### 3.3. Priorities Determined by the IPGA Model

Upon analyzing the gaps between people’s perceived importance and behavioral performance of health check-up preventive behavior in hospitals, 13 items had a statistically significant (*p* < 0.05) gap (Table 3). According to the IPGA model, nine items in Quadrant II needed urgent improvement (Figure 1). The first priority was the quality of a medical device, and the second priority was a wish to understand their health status. Other priorities in sequence were sufficient health knowledge, the completeness of items, being motivated by friends and family, process layout planning, a convenient location, expense needed, and previous experience (pleasant or unpleasant). 

### 3.4. Relationships between People’s Characteristics, Health Status, and Performance Gaps

To further evaluate the relationships between people’s characteristics, health status, and performance gaps, multiple regression analysis was performed. As shown in Table 4, the two factors of expense needed and process layout planning were negatively related to age, that is, younger people were more concerned about expenses and process layout planning when they attended health check-ups. Those with a college degree or above wished to understand their health status more than those with a high school education or below who attended health check-ups, and those with a master’s degree were more concerned about sufficient health knowledge and the quality of the medical device. Employed workers paid more attention to a convenient location, the completeness of items, and sufficient health knowledge than self-employed workers. Non-workers valued sufficient health knowledge more than self-employed workers. Those who had health screening experience were more interested in understanding their health status and more influenced by previous experience. Those who had no history of chronic diseases were more influenced by previous experience and more concerned about sufficient health knowledge.

## 4. Discussion

Previous studies based on communities have looked at many types of barriers to preventive care (i.e., why not do). This study provides an insight into the key factors and performance gaps (i.e., why do and how to do) of people who attended a hospital to undergo health check-ups, which may effectively provide strategies for improving the utilization of preventive care services.

Among the study participants in this study, people who attended individual private health check-ups and adult preventive health services voluntarily went to the hospital for health check-ups; they were mostly women who were married. In general, women have been found to have a higher utilization of preventive care services [37]. The possible explanations for this are that women are more sensitive and expressive of feelings than men and show a higher awareness of health problems and symptoms. Therefore, they often have worse perceptions about their own health and have a higher likelihood of using healthcare services [38]. Women were judged to pay more attention to their physical and mental conditions [39].

Seeking regular health check-ups is considered a self-care behavior. Individuals need to have cognitive abilities to learn, perceive, interpret, reason, and respond [40] and face multiple non-financial barriers to care, such as time pressure, complexities in the appointment process, the appropriate location [41], and employment precariousness [42]. Chien et al. [30] adopted a community-based cross-sectional study design in Taiwan and found that the factors that increased people’s willingness to participate in health screening were mainly determined by convenient locations and diagnostic facilities. In contrast, good health, a lack of time, long screening procedures, negative prior experience, and a lack of knowledge made people reluctant to participate. However, when people arrive at the hospital to undergo a health check-up, the most important factors are a wish to understand one’s health status, the quality of a medical device, the completeness of items, and process layout planning; this is the difference between the hospital-based study participants in this study and community-based study participants in previous studies.

An interesting aspect is how the gaps between barriers and drivers of preventive care services are bridged. Individuals’ intrinsic motivation to understand their health status may be the key factor; otherwise, even with available information and tools, their efforts will still be in vain [43]. One study showed that a higher purpose in life was related to a greater utilization of preventive care services and proactive preventive healthcare behavior [44]. Frankl said, “Those who have a ‘why’ to live, can bear with almost any ‘how’ [45].” This study showed that a wish to understand one’s health status, the quality of a medical device, the completeness of items, and process layout planning were the priorities in undergoing health check-ups, which clearly indicates that the participants had a higher awareness and a motivation for preventive healthcare. However, the factors of whether an incentive was offered (such as gifts and discounts) and whether somebody was asymptomatic without a check were not considered important enough to influence health check-up behavior.

As for the relationships between people’s characteristics and performance gaps, this study demonstrated that younger people were more concerned about expenses and process layout planning when they attended health check-ups. This is probably because young people take time off for health check-ups and thus have time pressures. They generally lack confidence in commercial health check-up packages [46] and are more likely to inquire about prices. People with a higher education attainment have a higher acceptance toward new health-related information and have better communication skills [47], so they want to know more about their health status and think that sufficient health knowledge and the completeness of items are very important. People who were not satisfied with the current service design were also found to be related to the willingness to continue to participate in health check-ups in the future [32], which is consistent with this study, which found that those with health screening experience and those without a history of chronic diseases were more influenced by previous experience; therefore, the concept of a service system design including the entire detailed process is a crucial factor that affects people’s decision to attend health check-ups [32].

The limitations of this study include the fact that participant inclusion was based on people who attended the hospital and were recruited with consent, so there may have been a selection bias. The study sample also came from a single regional hospital, so it may not be generalizable, and the study participants refer to the people who attended a hospital to undergo a health check-up, so any other preventive behaviors may not be generalizable either.

## 5. Conclusions

This study applied the IPGA approach for the first time to analyze the key factors and performance gaps in the behavior of people who attended a hospital to undergo a health check-up. Promoting the effectiveness of hospital performance may improve the positive cycle of people’s health check-up behavior from the community to the hospital. Continuing to promote a knowledge of preventive care services is necessary, and it is very important for people to have a good experience when undergoing health check-ups in the hospital. Therefore, for health check-ups in the hospital, meeting the person’s personalization needs, improving the quality of the medical devices and the completeness of items, and having appropriate process layout planning, a detailed interpretation of the results, and further follow-ups are important strategies that could improve the utilization of preventive care services in order to ensure personal health and well-being.

## Figures and Tables

**Figure 1 ijerph-19-13195-f001:**
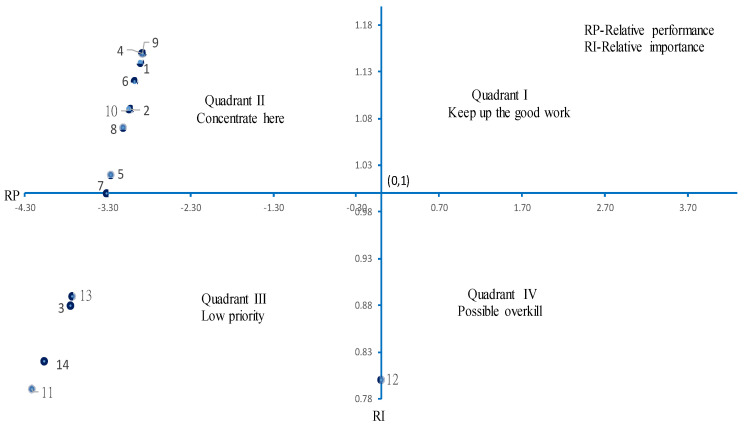
Results of the importance–performance gap analysis model.

**Table 1 ijerph-19-13195-t001:** Participant characteristics and comparative analysis.

	Employee Preventive Health Services	Personal Preventive Health Services	Adult Preventive Health Services	
	N %	N %	N %	*p* Value
Age Mean (SD)	39.9 (11.3)	45.5 (12.9)	52.2 (9.7)	0.000
Sex				0.000
Male	182 68.4	46 42.6	46 45.1	
Female	84 31.6	62 57.4	56 54.9	
Education				0.000
≤High school	17 6.4	22 20.5	42 41.2	
College	124 46.6	51 47.7	47 46.1	
≥Master	125 47.0	34 31.8	13 12.7	
Marital				0.000
Married	161 60.8	82 75.9	82 80.4	
Single	104 39.2	26 24.1	20 19.6	
Occupation				0.000
Self-employed	5 1.9	7 6.6	16 15.7	
Employed	261 98.1	72 67.9	43 42.2	
Non-worker	0 0.0	27 25.5	43 42.2	
Treatment experience				0.008
No	162 60.9	75 69.4	49 48.5	
Yes	104 39.1	33 30.6	52 51.5	
Health screening experience				0.398
No	125 47.0	42 39.3	44 44.9	
Yes	141 53.0	65 60.7	54 55.1	
Chronic disease history				0.003
No	238 89.5	87 80.6	77 76.2	
Yes	28 10.5	21 19.4	24 23.8	
BMI				0.373
Underweight	8 3.0	6 5.6	2 2.0	
Healthy weight	137 51.5	51 47.2	45 44.1	
Overweight	121 45.5	51 47.2	55 53.9	

**Table 2 ijerph-19-13195-t002:** Items affecting individuals’ utilization of health check-up programs: importance and influence on performance.

Items	Company Employee Health Check-Ups				Individual Private Health Check-Ups				Adult Preventive Health Services			
	Importance		Performance		Importance		Performance		Importance		Performance	
	Positive % ^a^	Ranking	Negative % ^b^	Ranking	Positive % ^a^	Ranking	Negative % ^b^	Ranking	Positive % ^a^	Ranking	Negative % ^b^	Ranking
1. sufficient health knowledge	95.1	1	56.3	5	93.5	3	63.2	5	95.1	1	44.1	6
2. motivated by friends and family	86.4	5	56.7	4	89.7	4	60.0	7	90.2	4	44.1	6
3. with a companion	45.7	11	23.2	13	55.1	10	42.5	10	41.2	12	16.7	13
4. wish to understand health status	95.1	1	68.8	1	95.3	1	74.3	1	92.2	3	48.0	3
5. previous experience	70.0	8	41.7	8	73.6	8	55.7	8	70.3	9	31.4	10
6. completeness of items	92.4	3	60.2	3	89.7	4	68.9	3	88.2	5	47.1	4
7. expense needed	63.3	9	39.4	9	69.8	9	54.7	9	70.6	8	35.3	8
8. convenient location	81.4	7	51.9	7	81.1	7	60.4	6	85.3	7	46.1	5
9. quality of medical device	92.4	3	67.4	2	94.4	2	73.6	2	93.1	2	63.7	1
10. process layout planning	83.3	6	52.3	6	86.0	6	64.2	4	86.3	6	52.5	2
11.incentive offered	27.5	14	17.4	14	34.6	13	22.6	14	27.5	14	14.9	14
12. asymptomatic without check	29.8	13	24.2	12	21.7	14	30.2	13	32.0	13	18.8	12
13. lack of time	48.9	10	34.8	10	45.3	11	36.8	11	50.0	10	34.3	9
14. afraid of potentially health problems	35.2	12	26.5	11	37.7	12	31.1	12	46.1	11	28.4	11

**^a^** Positive %, number of participants that answered “important” or “very important”/total number of participants. **^b^** Negative %, number of participants that answered “frequently influenced” or “always influenced”/total number of participants.

**Table 3 ijerph-19-13195-t003:** Results of the importance–performance gap analysis.

Items	t-Value	RP(*j*)	RI(*j*)	Quadrant	Distance D(*j*)	Priority
1. sufficient health knowledge	20.279 ***	−2.91	1.14	2	1.16	3
2. motivated by friends and family	16.747 ***	−3.03	1.09	2	0.96	5
3. with a companion	12.003 ***	−375	0.88	3	1.18	
4. wish to understand health status	15.554 ***	−2.89	1.15	2	1.20	2
5. previous experience	13.885 ***	−3.27	1.02	2	0.78	8
6. completeness of items	15.956 ***	−2.98	1.12	2	1.04	4
7. expense needed	12.029 ***	−3.31	1.00	2	0.78	8
8. convenient location	14.361 ***	−3.12	1.07	2	0.86	7
9. quality of medical device	15.067 ***	−2.88	1.15	2	1.22	1
10. process layout planning	15.032 ***	−3.05	1.09	2	0.94	6
11.incentive offered	8.021 ***	−4.22	0.79	3	1.73	
12. asymptomatic without check	1.694	0.00	0.80	3	1.36	
13. lack of time	6.884 ***	−3.73	0.89	3	1.15	
14. afraid of potentially health problems	4.779 ***	−4.07	0.82	3	1.56	

*** *p* < 0.001.

**Table 4 ijerph-19-13195-t004:** Regression analysis based on participant characteristics, health status, and the items in Quadrant II of the importance–performance gap analysis grid.

Independent Variables	Dependent Variables							
	Sufficient Health Knowledge	Understand Health Status	Previous Experience	Completeness of Items	Expense Needed	Convenient Location	Quality of Medical Device	Process Layout Planning
Age	−0.065	−0.059	0.014	−0.091	−0.0141 *	−0.102	−0.019	−0.178 *
Education								
High school (ref)								
College	0.075	0.210 **	0.009	0.040	0.184 *	0.098	0.085	0.093
≥Master	0.164 *	0.288 ***	0.072	0.148	0.122	0.111	0.166 *	0.066
Occupation								
Self-employed (ref)								
Employed	0.185 *	0.114	0.122	0.170 *	0.156	0.205 *	0.031	0.017
Non-worker	0.165 *	0.083	0.091	0.122	0.109	0.148	0.043	0.022
Health screening experience								
No (ref)								
Yes	0.068	0.129 **	0.160 **	0.093	0.027	0.031	0.093	0.037
Chronic disease history								
No (ref)								
Yes	−0.135 **	−0.085	−0.108 *	−0.079	0.046	−0.017	−0.055	0.054

******p* < 0.05, ******
*p* < 0.01, *******
*p* < 0.001. Note: the table shows significant items.

## Data Availability

The data were collected from the participant’s self-administered questionnaire with participant consent during the current study. The datasets are not publicly available but are accessible from the corresponding author upon reasonable request.

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
