# Peer review of "Strategies for Improving the Utilization of Preventive Care Services: Application of Importance–Performance Gap Analysis Method"

_ijerph, 2022, doi:10.3390/ijerph192013195_

Round 1

Reviewer 1 Report

The paper is interesting, and addresses an important topic - patient participation in preventive care. The manuscript has good potential, but there are some minor and major issues that must be clarified first. I include below a list of questions and recommendations. 

1. Abstract: although it conveys the main idea of the paper, I recommend a structured abstract, with clear specification of the aim, background, research gap, materials and method, results and implications. As it is now, it is now, it seems to be a long list of generalities that do not necessarily derive from your research. 

2. Section 2.2. You mention that the questionnaire was developed based on the extant literature, however no references are included. Please cite the appropriate literature that support your measurement instrument and explain what theoretical background and conceptual model you employed. 

3. No research hypotheses are formulated. Please reconsider this aspect. Build your hypotheses based on the existing literature and cite the appropriate references. 

4. On lines 193-195 you write: "For the 14 items of influencing factors perceived as important and having an influence on health check-up behaviors as seen by the people, Cronbach’s α values were all >0.78". What do you mean by this? How many Cronbach's Alpha values did you computed, corresponding to what groups of items, and what are their exact values?  

5. Please include an annex with the exact items used in your questionnaire

6. Table 4: you have some rows with entries "0". I assume that this is your way of indicating the reference category of your predictors. Please reconsider this notation - this is not how the estimated coefficients of a qualitative variable is reported. You only need to set the reference category as Reference, and report the other coefficient(s). It comes without saying that the coefficients of the other categories are interpreted relative to the reference one. Check some good quality papers reporting regression results and you will see what I mean. 

7. What software did you use in conducting data analysis? Please include this information in section 2.3. 

8. Regarding the IPGA tool: was this methodology used before in similar studies? Is this methodology better than other methodologies used in similar studies? Please explain and include references.

9. On lines 107-108 you write: "The questionnaire for this study was developed with reference to a literature review to collect the influencing factors related to health check-ups". This leads to the idea that the 14 factors included in your questionnaire are determinants of the preventive behavior, namely the health check-up. However, in Table 4 your approach is the other way around: health check-up is one of the predictors of the scores recorded to each of the 14 items. As a consequence, you are not exploring the natural relationship among your variables, and this messes up both your model and your argument. Please reconsider: this is one major issue of this manuscript. Many of the results and also your conclusions discuss health check-up as an outcome variable, whereas in Table 4 it's nothing but a predictor. 

10. Lines 301-305: you referred to the limitations of the study and I agree that the convenience sample that you used may influence the generality of the results. However, there are some other limitations that must be discussed. One of them refers to the fact that you target one very general preventive behavior, namely health check-ups. There are many other preventive behaviors out there and you don't refer to them. So please clarify that your results are valid only if this type of  preventive behavior is concerned. 

Reviewer 2 Report

The authors first review the literature on health maintenance and preventive checkups at the hospital level. This is well done, with a good English language style. Sound ideas are generated for a study of how 476 individuals referred for hospital checkups view the utility and importance of such checkups, and the statistical analyses assess the importance of 14 key reasons for responding to the checkup. The results offer important guides for future research, policy and practice in this area of health prevention and education. Obviously, prevention is better than cure, and programs such as this may have important economic implications.

Minor revisions are suggested: the Abstract should include more about the empirical work undertaken. It would also be useful to future researchers if the English and Chinese versions of the questionnaires were contained in an accessible downloadable file - this would enable future researchers to replicate this work.

Round 2

Reviewer 1 Report

The authors seem unwilling to respond to the major issue of the manuscript, raised on point 9 in my previous review recommendations. Unless this aspect is addressed I cannot recommend publication. In addition, the answer provided by the authors suggests that they are unaware of the main goal of a regression analysis - which is making predictions. 

I copy the recommendation below, once again:

On lines 107-108 you write: "The questionnaire for this study was developed with reference to a literature review to collect the influencing factors related to health check-ups". This leads to the idea that the 14 factors included in your questionnaire are determinants of the preventive behavior, namely the health check-up. However, in Table 4 your approach is the other way around: health check-up is one of the predictors of the scores recorded to each of the 14 items. As a consequence, you are not exploring the natural relationship among your variables, and this messes up both your model and your argument. Please reconsider: this is one major issue of this manuscript. Many of the results and also your conclusions discuss health check-up as an outcome variable, whereas in Table 4 it's nothing but a predictor. 
